# Pathophysiology of Angiotensin II-Mediated Hypertension, Cardiac Hypertrophy, and Failure: A Perspective from Macrophages

**DOI:** 10.3390/cells13232001

**Published:** 2024-12-04

**Authors:** Kelly Carter, Eshan Shah, Jessica Waite, Dhruv Rana, Zhi-Qing Zhao

**Affiliations:** Cardiovascular Research Laboratory, Mercer University School of Medicine, Savannah, GA 31404, USA

**Keywords:** angiotensin II, cardiac hypertrophy, hypertension, heart failure, macrophages, myocardial fibrosis

## Abstract

Heart failure is a complex syndrome characterized by cardiac hypertrophy, fibrosis, and diastolic/systolic dysfunction. These changes share many pathological features with significant inflammatory responses in the myocardium. Among the various regulatory systems that impact on these heterogeneous pathological processes, angiotensin II (Ang II)-activated macrophages play a pivotal role in the induction of subcellular defects and cardiac adverse remodeling during the progression of heart failure. Ang II stimulates macrophages via its AT1 receptor to release oxygen-free radicals, cytokines, chemokines, and other inflammatory mediators in the myocardium, and upregulates the expression of integrin adhesion molecules on both monocytes and endothelial cells, leading to monocyte-endothelial cell-cell interactions. The transendothelial migration of monocyte-derived macrophages exerts significant biological effects on the proliferation of fibroblasts, deposition of extracellular matrix proteins, induction of perivascular/interstitial fibrosis, and development of hypertension, cardiac hypertrophy and heart failure. Inhibition of macrophage activation using Ang II AT1 receptor antagonist or depletion of macrophages from the peripheral circulation has shown significant inhibitory effects on Ang II-induced vascular and myocardial injury. The purpose of this review is to discuss the current understanding in Ang II-induced maladaptive cardiac remodeling and dysfunction, particularly focusing on molecular signaling pathways involved in macrophages-mediated hypertension, cardiac hypertrophy, fibrosis, and failure. In addition, the challenges remained in translating these findings to the treatment of heart failure patients are also addressed.

## 1. Introduction

Cardiac injury is a complex structural and functional impairment of ventricular filling or ejection of blood in the heart, resulting from hypertension, myocardial infarction, coronary artery disease, valvular heart disease (regurgitation/stenosis), arrhythmia, and dilated cardiomyopathy. From the pathological viewpoint, these abnormalities are characterized by ventricular hypertrophy, fibrosis, diastolic, and/or systolic dysfunction. Dynamic pathophysiological compensations through maladaptive structural remodeling and ventricular chamber dilatation reduce cardiac output, and eventually lead to heart failure [1,2,3].

It has been well-demonstrated that angiotensin II (Ang II) plays a crucial role in the pathological development of systemic vascular resistance, hypertension, heart failure, and kidney disease over the last few decades [4]. The classical understanding of Ang II production is initiated via renin production. A reduction in blood pressure caused by cardiac and vascular injury reduces the renal blood flow and enhances the release of renin, a peptide produced primarily by the renal juxtaglomerular cells. Plasma renin then enzymatically cleaves angiotensinogen from the liver to Ang I, which is then converted to Ang II by the angiotensin-converting enzyme (ACE) found in the pulmonary endothelium. All biological effects of Ang II on the vascular endothelium and heart are mediated by binding to two G-protein coupled receptors, the AT1 and AT2 receptors. In systemic regulation, the activation of the AT1R generates vasoconstriction, stimulates aldosterone secretion from the adrenal cortex, promotes water reabsorption, and increases blood pressure. In local cardiovascular regulation, the stimulation of the AT1R elicits inflammatory response, macrophage migration, fibroblast proliferation, tissue fibrosis, and maladaptive remodeling. Accordingly, selective inhibition of the ACE or blockade of the AT1R has been conventionally used in the treatment of vascular and cardiac injuries in clinical settings [4,5,6,7]. We have recently reported that pharmacological inhibition of the AT1R with telmisartan and glucagon-like peptide-1 (GLP-1) agonist inhibits inflammation, fibroblast proliferation, hypertension, and myocardial fibrosis in the experimental setting of Ang II infusion [8,9].

Monocytes/macrophages are a type of white blood cells in the innate immune system and are found in essentially all tissues of the body. After the monocytes are recruited into cardiac tissue from the yolk sac and fetal liver, they differentiate into classically activated M1 macrophages and alternatively activated M2 macrophages according to their active states and gene-expression profiles. Both M1 macrophages and M2 macrophages are closely related to inflammatory responses. In a steady-state heart, the M1 macrophages play a beneficial role in maintaining cardiac homeostasis by regulating arterial tone and facilitating cardiac electrical conduction [10]. M1 macrophages are also pro-inflammatory phagocytic cells involved in the initial stages of inflammation. In response to inflammatory stimulation, systemically, M1 macrophages release abundant pro-inflammatory cytokines to cause vasodilation; where locally, M1 macrophages upregulate surface adhesion molecules on endothelial cells to induce monocyte extravasation. In contrast, M2 macrophages are anti-inflammatory cells and are implicated in the resolution of the inflammatory process. These macrophages eliminate apoptotic cells, promote angiogenesis and neovascularization, and facilitate stromal cell activation and tissue repair [10,11]. A specific subset of macrophages called monocytes-derived macrophages (e.g., CCR2^+^ cells) from the bone marrow and spleen have been shown in the participation of inflammation, proliferation, and maturation, acting as M1-like resident macrophages during tissue remodeling [12,13]. These macrophages enhance cardiac hypertrophy and stimulate myocardial fibrosis by producing transforming growth factor beta 1 (TGF-β1), proliferating fibroblasts, activating Smad2/3 signaling, and promoting myocardial fibrosis [14]. In this regard, we have shown that Ang II significantly drives monocyte/macrophage recruitment via activation of the AT1R, and that the reduction in the recruitment of macrophages from the spleen by splenectomy or the AT1R blockade attenuates Ang II-induced maladaptive vascular and myocardial remodeling responses [15].

Here, we describe the phenotypes of tissue-resident macrophages and monocyte-derived macrophages in the cardiovascular system, define their physiological role in tissue homeostasis, and discuss their pathological role in the development of hypertension, hypertrophy, tissue fibrosis, and heart failure, primarily in the context of Ang II-mediated remodeling signaling pathways. We summarize promising therapeutic approaches on the attenuation of macrophage-mediated inflammation and tissue remodeling in the cardiovascular system. Finally, we discuss the future directions for the investigation of potential molecular mechanisms of macrophage inhibition and cardiovascular protection in general.

## 2. Origins and Phenotypes of Tissue-Resident Macrophages and Monocyte-Derived Macrophages

Macrophages play multifaceted roles in immune response and tissue homeostasis. In response to inflammation or infection, macrophages engulf foreign invaders, degrade dead cells, and promote tissue homeostasis by releasing multiple cytokines to coordinate immune response and inflammatory regulation. On the other hand, macrophages also lead to maladaptive regeneration, repair, and tissue fibrosis. There are two types of macrophage phenotypes: resident macrophages and monocyte-derived macrophages.

### 2.1. Tissue-Resident Macrophages

It is well-established that tissue-resident macrophages in most tissues during the prenatal period play a critical role in maintaining tissue homeostasis and appear as self-renewal rather than the replenishment with monocytes throughout the life span. They originate from the hematopoietic stem cells in the yolk sac and fetal liver during primitive hematopoiesis. In steady state conditions, resident macrophages proliferate in situ at low levels, but their proliferation rate increases dramatically during inflammatory stimulation [12,16]. These cells develop through a series of self-renewal stages including the common myeloid progenitor, the granulocyte-macrophage progenitor, the common macrophage, the dendritic cell precursor, the committed monocyte progenitor, and the tissue-resident macrophages [17]. Resident macrophages are heterogeneous cells that are found in virtually all tissues of adult mammals, where they can represent up to 10–15% of the total cell number in quiescent conditions. These macrophages allocate to different organs throughout the body and are named Kupffer cells in the liver, microglia in the brain, alveolar macrophages in the lung, histiocytes in the connective tissue, osteoMacs in the bone, langerhans cells in the skin, resident macrophages in the intestinal mucosa, and mesangium in the kidney (Figure 1A). Besides phagocytosis, they maintain tissue homeostasis and initiate specific adaptive immunity by recruiting other immune cells such as T lymphocytes [18]. Conventionally, during their development and in response to different microenvironmental stimuli, these macrophages undergo phenotypical polarization and functionally are classified as M1-like and M2-like phenotypes [17]. Cytokines produced from these cells are potent signaling molecules involved in chemotaxis, angiogenesis, tissue remodeling, and repair. When stimulated, M1 macrophages produce pro-inflammatory cytokines including interleukin-1β (IL-1β), IL-6, IL-12, IL-23, and tumor necrosis factor-α (TNF-α), whereas M2 macrophages produce anti-inflammatory cytokines such as IL-4, IL-13, IL-10, IL-1RA, and TGF-β.

In steady-state and non-inflammatory conditions, few macrophages are detected in the heart. In human observation, according to the expression of the surface markers CD14 and CD16, about 90% of classical M1 monocytes are defined as high CD14 but no CD16 expression (CD14^++^CD16^−^ or CD14^+^CD16^−^), whereas the other 10% of non-classical M2 monocytes are constituted with high CD16 and relatively lower CD14 expression (CD14^+^CD16^++^ or CD14^dim^CD16^+^) as well as the intermediate subset with low CD16 and high CD14 (CD14^++^CD16^+^ or CD14^+^CD16^+^) [17,19,20]. In animal studies, it was shown that most cardiac resident macrophages derive from the precursor of embryonic origin and are not replenished by circulating blood monocytes. Based on the expression of major histocompatibility complex II (MHC-II) and the C-C chemokine receptor 2 (CCR2), cardiac resident macrophages can be categorized into two distinguished separate and discrete subsets: MHC-II^low^CCR2^−^ and MHC-II^high^CCR2^−^ cells, which originated from yolk sac progenitors. They function to promote coronary development and cardiac regeneration and to facilitate electrical conduction within the atrioventricular node [21,22]. The macrophages expressing MHC-II^high^CCR2^+^ cells are mainly derived from blood monocyte recruitment. The expression of lymphocyte antigen 6 complex (Ly6C) can be used to distinguish inflammatory and reparative macrophages [20,23]. Ly6C^low^ macrophages, generally regarded as tissue-resident macrophages, are considered to be derived from the embryonic yolk sac, whereas Ly6C^high^ macrophages are thought to be derived from bone marrow-derived monocytes. Furthermore, cardiac resident macrophages retain a high level in the expression of chemokine (C-X3-C motif) ligand 1 (CX3CL1) and react to chemokine receptor 1 (CX3CR1). The presence of these two key receptors, including the pro-inflammatory CCR2 and the anti-inflammatory CX3CR1, builds the basis of the homing mechanism of Ly6C^low^ resident macrophages and Ly6C^high^ monocyte-derived macrophages at inflammatory sites. In the heart, resident macrophages proliferate at low levels in homeostatic conditions, and play a beneficial role in maintaining cardiac electrical conduction and arterial tone regulation and stimulating cardiomyocyte proliferation and angiogenesis [24,25,26].

### 2.2. Monocyte-Derived Macrophages

Hematopoiesis passes from the yolk sac and fetal liver to the bone marrow during the postnatal period (Figure 1B). The self-renewal of cardiac resident macrophages declines with aging, and these cells can be replenished by monocyte-derived macrophages, which are a group of cells circulating in the blood, bone marrow, and spleen. Macrophages originate from hematopoietic stem cells (HSCs) as a common myeloid-monocytic progenitor, i.e., the colony-forming unit of granulocytes and monocytes. Subsequently, these progenitors progressively initiate the myeloid and monocytic lineage through a series of sequential differentiation stages, and eventually migrate into tissues where they accomplish their terminal differentiation into monocyte-derived macrophages.

In animals, according to the expression of Ly6C, monocytes can be divided into Ly6C^high^ monocyte and Ly6C^low^ monocyte. Under the action of chemokine (CC-motif) ligand 2 (CCL2), also known as monocyte chemoattractant protein-1 (MCP-1), during inflammation, Ly6C^high^ monocytes, mainly derived from circulating monocytes, infiltrate to the damaged myocardium, where they are differentiated into recruited CCR2^+^ macrophages (herein called Ly6C^high^CCR2^+^ cells) to release pro-inflammatory cytokines and pro-fibrotic mediators (Figure 1B). In contrast, Ly6C^Low^ monocytes (CCR2^−^) are shown more regenerative. The circulating monocyte expressing CD14^+^CD16^−^ and CD14^−^CD16^+^ phenotypes in humans are functionally matched with Ly6C^high^ and Ly6C^low^ cells in murine, respectively [19,27].

Endogenous chemokine ligands, when binding to their corresponding cell-surface G-protein-coupled receptors, control the migration and trafficking of monocytes in response to inflammation. As summarized in Table 1, chemokine receptors display different levels of binding specificity and promiscuity to their ligands. For example, CCL ligands bind to CCR receptors, CXCLs bind to CXCR receptors, and CX3CLs bind to CX3CR receptors. The CCL2-CCR2 axis has been shown to be the predominant ligand/receptor regulator in association with monocyte/macrophage chemotaxis [28].

Both types of macrophages are present in the adult heart, including yolk sac-derived CCR2^−^ macrophages and bone marrow progenitors-derived CCR2^+^ macrophages. CCR2^−^ macrophages express higher amounts of CD163, CD206, and neurophilin, and exert anti-inflammatory and pro-fibrotic actions, whereas CCR2^+^ macrophages display a higher expression of markers CD38, CD274, CD197, CD54, CD82, CD86, and Slamf7, and are mainly involved in pro-inflammatory activation response, as deeply reviewed in [28,29,30]. It has been reported that monocyte-derived cardiac macrophages through phenotypic switching change increase with age from CCR2^−^ macrophages to CCR2^+^ macrophages. CCR2^+^ macrophages promote inflammation at the early stage of wound healing, and CCR2^−^ macrophages inhibit inflammation during the later stage. When the CCR2^−^ phenotype is predominantly replaced by CCR2^+^ monocyte-derived macrophages, the unbalanced CCR2^−^ to CCR2^+^ ratio results in chronic inflammation and tissue fibrosis [31,32].

### 2.3. Role of the Spleen in Ang II-Activated Ly6C^high^CCR2^+^ Monocytes

Like the bone marrow, the spleen acts as a reservoir for Ly6C^high^CCR2^+^ monocytes and participates in the repopulation of cardiac monocyte-derived macrophages. Ly6C^high^CCR2^+^ monocytes undergo Ang II–dependent mobilization and migrate into the cardiac tissue, as identified in pathological conditions of atherosclerosis, heart failure, and stroke [33,34]. In the mouse model of permanent coronary ligation, ACE inhibitor enalapril arrested the release of monocytes from the splenic reservoir, and consequently reduced their recruitment into the infarct zone, accompanied by improved cardiac function. In vitro migration assays showed that this process is mediated by inhibiting AT1R signaling. These data suggest that the release of monocytes from the splenic reservoir after myocardial infarction can partially attribute to Ang II-elicited inflammation. Splenectomy reproduced anti-inflammatory effects of enalapril on splenic monocyte mobilization [34]. In coincidence with these findings, we have previously reported that the splenic release of monocytes participates in Ang II-elicited cardiac fibrosis and hypertension by modulating macrophage population in the myocardium and blood vessels. Treatment with the AT1 receptor blocker telmisartan or splenectomy reduced the recruitment of monocyte-derived macrophages, downregulated the expression of MCP-1 and TGF-β1, and attenuated the deposition of collagens and interstitial/perivascular fibrosis, further confirming a pathological role of cardiosplenic axis in Ang II-mediated cardiac injury [14,15]. These data were consistent with other reports showing that cancer growth is associated with Ang II-enhanced macrophage production from the spleen’s red pulp, and the blockade of the AT1R attenuates sepsis-induced cardiomyopathy by regulating macrophage polarization from the spleen [35,36].

## 3. Ang II-Modulated Adhesion Molecule Expression on Monocytes and the Vascular Endothelium

The extravasation of monocytes across the vascular endothelium is crucial for evoking inflammatory responses and fibrotic cascades in the myocardium. The adhesion of monocytes to the vascular endothelium and their subsequent migration into the interstitial region is a hallmark of cardiac fibrosis. The initiation of cell-cell interactions between monocytes and endothelial cells depends on adhesion molecules expressed on both types of cells. These processes are influenced by many humoral factors, of which Ang II seems to be critical. Besides being a potent vasoactive peptide, Ang II exerts pro-inflammatory effects on the monocyte and vasculature by upregulating adhesion molecules and promoting cell-cell interactions.

### 3.1. Monocytes and Endothelial Cells Interactions

The migration of monocytes from the circulation to peripheral organs involves sequential cell-cell interactions between monocytes and endothelial cells. The classical cascade of monocyte recruitment is defined by the following steps: capture, rolling, activation, firm adhesion, crawling and transendothelial migration. Monocyte extravasation to inflammatory sites is a multi-step process involving many adhesion molecules expressed on monocytes and endothelial cells. The initial capture of monocytes along the inflamed endothelium relay on both P-selectin glycoprotein ligand-1 (PSGL-1) and lectin-like molecules (L-selectin) expressed on monocytes as well as P-selectin and E-selectin expressed on endothelial cells [37,38,39]. The rolling mediated by these selectins allows the monocytes to interact with other adhesion molecules through weak adhesion (low-affinity binding) on the endothelium. The subsequent firm adhesion and crawling occur mainly through the Ig-supergene family of glycoproteins, in which macrophage-antigen-1 (Mac-1) and lymphocyte function-associated antigen 1 (LFA-1) bind to their cognate receptors: intercellular adhesion molecule-1 and -II (ICAM-1, ICAM-2), respectively. In parallel to this adhesive complex, the neutrophil β1-integrin very late antigen-4 (VLA-4) and its endothelial binding partner vascular cell adhesion molecule-1 (VCAM-1) can further enhance monocyte-endothelium interactions [40,41]. Diapedesis of monocytes occurs via upregulating the platelet endothelial cell adhesion molecule (PECAM1) and vascular endothelial (VE)-cadherin at junctions between the endothelial cells. After the monocytes penetrate through the endothelial cells, they migrate into the extracellular matrix of the tissues, where they differentiate into monocyte-derived macrophages to release multiple cytokines and chemokines in response to Ang II stimulation (Table 1, Figure 2).

The interactions between monocytes and endothelial cells are facilitated by cytokines and chemokines released from monocytes and endothelial cells. Pro-inflammatory cytokines including TNF-α, iNOS, IL-1β, IL-6, IL-12, and IL-23 trigger the production of monocyte-attracting chemokines CCL2, CXCLs, and CX3CLs, and augment the expression of cell-adhesion molecules [42,43]. Chemokine ligands bind to their corresponding cell-surface specific chemokine receptors to exert their biological effects. These receptors are G-protein-coupled signal via pertussis toxin-sensitive Gi proteins. Approximately 50 endogenous chemokine ligands have been identified [28]. Among these chemokine ligands and receptors, CCL2 and CCL7 and their receptor CCR2 and CCR1 have been reported as the key chemokines to facilitate the transmigration of Ly6C^high^CCR2^+^ monocytes into peripheral tissues. This in turn increases the diapedesis of CCL2-CCR2 axis-dependent monocytes to the inflammatory sites [44,45]. The activation of CCL2-CCR2 axis also alters the conformation of VLA-4, leading to higher-affinity interaction with its receptor VCAM-1. On the apical surface of endothelial cells, the bound chemokine MCP-1 and macrophage inflammatory protein-1 (MIP-1) can activate monocyte β-integrins for tight adhesion to ICAM-1. On the other hand, Ly6C^low^CCR2^−^ monocytes bind to the endothelium via CCL3-CX3CR1, and transmigrate in an LFA1/ICAM1-dependent manner for the normal function of monocytes [46,47].

### 3.2. Augmentation in the Extravasation of Monocyte-Derived Macrophages by Ang II

Ang II regulates multiple steps in leukocytes/monocytes recruitment into the vessel wall and the myocardium. Via activating the AT1R, Ang II stimulates the production of pro-inflammatory cytokines and chemokines, enhances the adhesion of leukocytes/monocytes to endothelial cells by upregulating cell-adhesion molecules, and promotes monocyte transendothelial migration (Figure 2). Data from our laboratory and others have shown that Ang II stimulates nicotinamide adenine dinucleotide phosphate (NADPH) oxidase, a major extracellular enzymatic source of ROS, activates many downstream signaling targets such as MAP kinases, transcription factors, protein tyrosine phosphatases, and tyrosine kinases, and further influences the production of chemokines and the expression of adhesion molecules [9,48,49]. In the rat model of Ang II osmotic infusion, we have recently reported that Ang II upregulates the expression of primary NADPH oxidase NOX4 isoform and the production of ROS via acting the AT1R. The increased ROS and oxidative stress by NOX4 in turn cause endothelial NO synthase (eNOS) uncoupling, enhance the expression of ICAM-1 and MCP-1, and facilitate the migration of monocytes to the myocardium. Furthermore, NOX4-mediated ROS generation triggers the vicious cycle of mitochondrial ROS generation and leads to mitochondrial dysfunction [9].

In the mice model of intraperitoneal Ang II injection, Ang II significantly enhanced PSGL-1 and P-selectin-mediated monocyte-endothelial cell interactions. Upregulated PSGL-1 and P-selectin on monocytes and endothelial cells were associated with the production of endothelial NADPH oxidase-derived ROS. In endothelial cells, Ang II upregulates the expression of E-selectin, VCAM-1, and ICAM-1, and leads to the release of a myriad of cytokines, chemokines, matrix metalloproteinases, and vasoactive agents [50,51]. In addition, Ang II promotes the recruitment of monocyte-derived CCR2^+^ macrophages by upregulating MCP-1 binding to its receptor CCR2^+^, and results in alterations in vascular function, extracellular matrix turnover, and tissue fibrosis [52,53]. Blocking the AT1R with candesartan or losartan, and neutralizing PSGL-1 or P-selectin using antibodies, inhibited the rolling and adhesion of monocytes on cerebral venules, suggesting that the migration of monocyte-derived macrophages relies on the AT1R and adhesion molecules, which are present on both monocytes and endothelial cells. Therefore, blocking the AT1R not only inhibits inflammation, ROS production, monocyte adhesion, and migration, but also attenuates vascular remodeling and subsequent vascular dysfunction [54,55].

## 4. Roles of Ang II-Elicited Vascular Histopathological Alterations and CCR2^+^ Macrophage Vascular Infiltration in the Development of Hypertension

Uncontrolled hypertension is the most common risk factor for the development of heart failure, aortic aneurysm, angina pectoris, atrial fibrillation, chronic kidney disease, and stroke. The evolution of hypertension is determined in large part by reduced vascular diameter due to increased vascular contraction and arterial remodeling [56,57]. Throughout the progression of hypertension, Ang II elicits vascular inflammation and enhances the interactions between the monocyte/macrophage and the arterial wall, leading to vascular adverse remodeling and hypertension.

### 4.1. Characteristics of Vascular Morphology, Function, and Inflammation Under Ang II Stimulation

The arterial wall consists of three distinct layers: tunica intima (endothelial cells), tunica media (smooth muscle cells), and tunica adventitia (tunica externa and extracellular matrix scaffold). Each layer exhibits specific histologic, biochemical, and functional characteristics (Figure 3). The tunica intima consists of a single layer of longitudinally oriented endothelial cells that is the innermost layer of any vessel. These cells secrete bioactive substances to modulate the vascular tone, control the platelet/monocyte adhesion, and prevent thrombotic formation. The tunica media is the thickest middlemost layer of the vessel wall and consists of smooth muscle cells that are primarily made up of elastic fibers. Contraction and relaxation of the circular smooth muscles decrease or increase the diameter of the vessel lumen, respectively. Specifically, vasoconstriction decreases blood flow, and vasodilation increases blood flow. The tunica adventitia is the outermost layer of the vessel wall that is the interface between the vessel wall and its neighboring tissues. It contains many different interacting cell types, including macrophages, fibroblasts, T cells, B cells, mast cells, and dendritic cells to carry out innate immune functions in response to inflammation. The vasa vasorum are specialized vessels found in the adventitia of the vessel wall. The opening of the vasa vasorum is associated with transporting molecules and cells from the blood to the adventitia. The neovascularization of the vasa vasorum is induced by stress and inflammation, but not stimulated by change in the vessel wall thickness [58,59]. The adventitia works as a dynamic compartment for the trafficking and migration of monocytes. Two mechanisms in response to vascular inflammation have been proposed: the “inside-out” response is initiated from injured intima of blood vessels by upregulating surface adhesion molecules and inflammatory mediators, and therefore, enhances the monocyte homing and the transmigration into the intima and/or media; the “outside-in” hypothesis states that vascular injuries are initiated and perpetuated from the adventitia, and leads to vascular medial and intimal remodeling [59]. Therefore, the communications among the intima, media, and adventitia corporately determine the homeostasis of blood vessels and the progression and/or regression of vascular disease in physiological and hypertensive conditions.

### 4.2. Ang II and Inflammatory Mediators

Among multiple triggers involved in the production of hypertension, including the excessive production of ROS, the vascular morphological alteration-induced progression of atherosclerosis and the abnormalities of the sympathetic nervous system, it has been generally accepted that Ang II is the most critical player in the development of long-term hypertension. In physiological conditions, Ang II regulates extracellular volume and blood pressure by secreting aldosterone from the adrenal cortex in the zona glomerulosa, stimulating the Na-H antiporter in the proximal convoluted tubule to increase sodium reabsorption, contracting the vascular smooth muscle in the arterioles, increasing sympathetic outflow from the central nervous system, and releasing vasopressin from the hypothalamus. On the other hand, Ang II-induced functional changes in the tunica intima are critical in the development of hypertension [48,51].

Ang II increases the generation of ROS, chemokines, and cytokines, and results in the reduction in the bioavailability of eNOS, which are responsible for inflammation, vascular remodeling, vasoconstriction, and perivascular fibrosis. As shown in Figure 3, the activation of the AT1R enhances the expression of NADPH oxidase, a major extracellular enzymatic source of vascular ROS, and induces the uncoupling of eNOS on endothelial cells. We recently reported that Ang II infusion for 4 weeks in rats upregulates the expression of NOX4, ICAM-1, and MCP-1, and downregulates the expression of eNOS on the aortic endothelium, in company with significantly increased aortic wall thickness, mitochondrial dysfunction, and perivascular fibrosis. The activation of NOX4 in this study was associated with the downregulation of SIRT-3 (NAD-dependent deacetylase sirtuin-3), which leads to the reduction in respiration and the production of ROS. In addition, NOX4 also upregulated Bnip3 (Bcl-2/adenovirus E1B 19-kDa-interacting protein 3) to induce mitochondrial swelling and respiratory collapse, the loss of mitochondrial membrane potential, the opening of the mitochondrial permeability transition pore, and the release of cytochrome C [9]. Therefore, Ang II enhances the expression of endothelial adhesive molecules and promotes endothelial dysfunction by activating NADPH/ROS and enhancing the specific release of cytokines/chemokines from migrated macrophages (Figure 2).

Increasing evidence proposes that the activation of toll-like receptor 4 (TLR4) is closely involved in the inflammatory process and hypertension. TLR4 initiates the down-stream signaling cascades via MyD88-mediated pathways and results in the activation of NF-κB. The subsequent transfers of activated NF-κB to the nucleus induces the transcription and translation of related inflammatory mediator genes, and thereby causes the release of pro-inflammatory cytokines and chemokines such as, IL-6, IL-8, IL-18, MCP-1, and TNF-α. Subsequently, NF-κB activation in endothelial cells results in the synthesis of adhesion molecules, enhances the extravasation of circulating monocytes into the vascular walls, and causes the propagation of inflammation in vascular remodeling [60,61]. We and others have shown that systemic infusion of Ang II upregulates NF-κB expression in endothelial cells of the aorta. Treatment with a neutralizing anti-TLR4 antibody, or an AT1R blocker, decreased inflammatory responses and NF-κB activity, suggesting that Ang II-induced vascular inflammation is mediated by AT1R and TLR4 on the endothelium [9,14,62].

### 4.3. Ang II Induces Vascular Hyperplasia and Hypertrophy

Vascular smooth muscle cells (VSMC, tunica media) play a pivotal role in regulating blood flow and maintaining blood pressure via facilitating the contraction and dilation of the vasculature. Multiple pro-hypertensive stimuli are involved in the regulation of VSMC function, such as the activation of the sympathetic nervous system, hemodynamic alterations, mechanical forces, and oxidative stress. Ang II-induced hypertrophy in VSMCs has been shown to play an essential role in eliciting vascular structural and functional changes in hypertension through inflammation, calcification, vasoconstriction, and hyperplasia. Multiple signal-transduction pathways have been implicated in the AT1R-modulated VSMC pathophysiological responses, such as matrix production, hypertrophy, hypercontractility, and vascular remodeling. Previous studies have reported that in cultured rat aortic VSMCs, Ang II stimulates both hyperplasia and cell proliferation by activating AT1R [63,64]. The AT1R is a prototypical G-protein-coupled receptor (GPCR) family to regulate diverse pathological processes through the different G protein isoforms [65,66]. The downstream signals of the AT1R stimulation include the activation of phospholipase C (PLC), phospholipase D (PLD), phospholipase A_2_ (PLA_2_), protein kinase C (PKC)-δ, p38 of MAPK family, platelet-derived growth factor receptor (PDGF), epidermal growth factor receptor (EGFR), and insulin-like growth factor receptor (IGFR). In addition, the activation of many intracellular non-receptor tyrosine kinases by Ang II, such as Src family kinases, Ca^2+^-dependent tyrosine kinases (e.g., Pyk2), p130Cas, FAK, phosphatidylinositol 3-kinase (PI3K), and JAK2/TYK2 (Janus kinase), is also associated with altered VSMC function in hypertension [66]. Alterations of these highly regulated signaling pathways in VSMC may associate with structural and functional abnormalities underlying VSMC oxidation, proliferation, and pathophysiological alterations such as vascular stiffness, compliance, hyperplasia, and remodeling, indicative of hypertension. From a molecular, cellular, and vascular perspective, this being said, there are many similarities, and perhaps too many opportunities for crosstalk between inflammatory stimuli and hypertensive outcomes [63,66].

### 4.4. Ang II Stimulates CCR2^+^ Macrophage Infiltration and Perivascular Fibrosis

Although the dysfunction of endothelial cells and VSMCs is a major trigger in the development of hypertension, abundant evidence has shown that perivascular fibrosis is associated with Ang II-induced hypertension. The tunica adventitia is in close contact with perivascular tissue that contains macrophages, fibroblasts, adipocytes, lymphatic vessels, nerves, and stromal cells, exhibiting mesenchymal stem cell–like properties [67,68]. They are linked by microvessels to regulate vascular physiology, homeostasis, and structural remodeling, in which they exert major influences on the progression of hypertensive vascular disease. Under the influence of Ang II-induced ROS, the interaction between adventitia and perivascular tissue through increased vasa vasorum neovascularization promotes the migration of monocytes/macrophages and the phenotypic switch of adventitial fibroblasts into migratory myofibroblasts toward the tunica media. This “outside-in” injury mechanism eventually leads to robust collagen deposition and perivascular fibrosis and increases the arterial wall stiffness and vascular resistance via inward remodeling [69]. Previous studies have shown that along with the production of inflammatory cytokines and the upregulation of adhesion molecules in the aortic wall, Ang II infusion resulted in Ly6C^high^CCR2^+^ macrophage infiltration from the aortic adventitia to the media. Using CCR2^+^ antagonist INCB3344 after the commencement of Ang II infusion significantly reduced the number of CCR2^+^ macrophages and limited aortic hypertrophy, providing the first direct experimental evidence showing a role of CCR2^+^ macrophages in adverse structural remodeling of the aorta [70]. Consistent with these findings, we have also found that Ang II infusion causes morphological and structural changes in the aortic wall, including a significant increase in positive staining of AT1R expression, macrophage accumulation, myofibroblast differentiation, perivascular fibrosis, and a thicker arterial wall, in coincidence with a reduction in the expression of eNOS and an elevation in blood pressure [15].

In summary, macrophages appearing in the arterial wall come from two different sources. In a steady state, intravascular macrophages are recruited at birth from the closing ductus arteriosus, and share the luminal surface with the endothelium, becoming interwoven in the tunica intima. These tissue-resident macrophages are found to regulate the growth of endothelial cells and VSMCs and maintain blood vessel hemostasis [71]. However, in the progression of Ang II-mediated hypertension, locally produced CCL2 from endothelial cells binds CCR2^+^ on monocytes through an “inside-out” mechanism that promotes the migration of circulating monocytes to the tunica media [72]. On the other hand, CCR2^+^ macrophages accumulated in intima give rise to local inflammation by secreting various pro-inflammatory cytokines and chemokines, leading to additional monocyte recruitments on the tunica media from the aortic adventitia and perivascular tissue through the “outside-in” mechanism. The disruption of the media immune privilege manifests as the loss of VSMCs, the destruction of the ECM architecture, and more intense monocyte infiltration [67,68,69].

Fibroblasts, the most abundant cell type within the tunica adventitia, play a critical role in the regulation of vascular wall function. In the normotensive state, fibroblasts organize and remove extracellular matrix through the production of soluble mediators such as cytokines, growth factors, and metabolites. However, in response to Ang II-induced hypertension, adventitial fibroblasts are activated, undergo a dynamic phenotypic change, and move into the tunica media. Multiple growth factors, receptors, or TGF-β1, released from the activated fibroblasts themselves, act as the most important signal transmitters in the determination of perivascular fibrosis [73,74], where the proliferation, differentiation, and production of extracellular matrix proteins occur. The proliferated myofibroblasts in VSMCs secrete a number of vasoactive substances, such as alpha smooth muscle actin (αSMA), TGF-β1, periostin, cartilage oligomeric matrix protein (Comp), connective tissue growth factor (CTGF), and collagen I, which, collectively, affect the medial tone of VSMCs and eventually result in vascular remodeling and hypertension [75,76].

## 5. Cardiac Hypertrophy/Fibrosis and CCR2^+^/CCR2^−^ Macrophages

Cardiac hypertrophy occurs in two conditions, either physiological or pathological. Physiological hypertrophy is characterized by normal morphological structure and enhanced cardiac function, whereas pathological hypertrophy is associated with an increase in the size of cardiomyocytes in response to pressure/volume overload and is characterized by cardiac fibrosis and dysfunction. CCR2^+^/CCR2^−^ macrophages have been regarded as one of the major cellular components of the heart to modulate physiological and pathological hypertrophy, whereas either they are reduced or augmented in the production of tissue fibrosis, particularly during Ang II stimulation.

### 5.1. Physiological Hypertrophy and Pathological Hypertrophy

Cardiac hypertrophy is classified into various variants depending on the geometries of the heart, and it is generally divided into two categories: physiological and pathological hypertrophy, which are associated with an increase in left ventricular wall thickness or volume load. Physiologic hypertrophy usually occurs in athletes or during pregnancy to normally adapt to a chronic pressure or volume overload. It is characterized with normal cardiac architecture and enhanced contractile function, typically expressed as a 5–10% increase in heart mass normalized to body mass. Pathological hypertrophy is associated with morphological changes (fibrosis and cell death) and reduced diastolic and/or systolic function [77,78,79]. From the perspective of the heart geometry, cardiac hypertrophy can also be classified into concentric hypertrophy and eccentric hypertrophy. Concentric hypertrophy is expressed as an increase in ventricular free wall thickness more than in length of sarcomeres with reduced ventricular dimension, leading to pressure overload and reduced filling in the left ventricle. However, eccentric hypertrophy is characterized as an increase in ventricular volume with the addition of sarcomeres in series, causing increased ventricular dimension and dilation [80,81]. Both concentric and eccentric hypertrophy can be detected in pathological hypertrophy, whereas concentric hypertrophy primarily appearing in physiological conditions is not necessarily pathological and can even be beneficial (Figure 4).

The development of pathological concentric and eccentric hypertrophy is associated with a chronic pressure and/or volume overload due to abnormal hemodynamic stress initiated by hypertension, myocardial infarction, dilated cardiomyopathy, valvular diseases, and congenital heart defects [77,78]. Many molecular and cellular signaling mechanisms have been proposed in the development of cardiac pathological hypertrophy. As it has been well-reviewed previously, the upstream signals activated by stretch are thought to occur through a number of signaling events involving stretch-activated nonselective cation channels and the production of growth hormones, such as thyroid hormone, insulin, insulin-like growth factor 1 (IGF-1), vascular endothelial growth factor (VEGF), cytokines (TNF-α, IL-1β, IL-6), and focal adhesion kinase [82,83,84,85]. The downstream targets identified include PI3K, AKT, ERK1/2, mammalian target of rapamycin (mTOR), mitogen-activated protein kinase, Ca^2+^/calmodulin-dependent protein kinase II, and protein kinase C [86,87,88]. We and others have previously reported that maladaptive cardiac remodeling underlying the disruption of collagen fibers, the accumulation of extracellular matrix (ECM), and the induction of interstitial fibrosis is a major contributor in the transition of pathological concentric to eccentric hypertrophy [8,89]. An increase in myocardial wall stiffness elicited by this maladaptive remodeling causes the pressure overload and poor ventricular filling during diastole. Over time, increased ECM proteolytic activity can result in the reduction of ECM content, ventricular dilatation, and volume overload, which is unable to effectively pump blood and leads to systolic dysfunction.

### 5.2. Roles of CCR2^−^ Macrophages and Ang II-Activated CCR2^+^ Macrophages in the Induction of Cardiac Pathological Hypertrophy

Ang II-elicited maladaptive remodeling involves the endothelium, VSMCs, cardiomyocytes, and inflammatory cells (e.g., macrophages and fibroblasts) in the ECM. As discussed above, the stimulation of the AT1R by Ang II increases the release of cytokines, IL-6, and NOXs from endothelial cells, results in the production of vasoconstrictor agents from VSMCs, and further leads to vascular remodeling (e.g., aortic wall thickening) and hypertension (e.g., pressure overload). Although the exact mechanisms of action in the transition of Ang II-induced concentric hypertrophy to eccentric hypertrophy are not fully understood, it appears that the degree in the production of cardiac perivascular and interstitial fibrosis may be one of the most important pathogenic factors. In this regard, the recruitment of monocytes-derived macrophages via the CCL2/CCR2 pathway has been proposed to be involved in collagen deposition and cardiomyocyte hypertrophy [90,91,92]. We found that soluble substances released from monocytes-derived macrophages, such as TGF-β1, proliferate fibroblasts to myofibroblasts and result in the production of collagens via Smads-mediated signaling pathways. Upon activation of the cellular surface TGF-β1 receptor on myofibroblasts by Ang II, intracellular phosphorylated Smad2/3 can form a heterotrimeric complex with Smad4 and subsequently bind to TGF-β1-targeted genes in the nucleus to synthesize ECM proteins. In the cardiac vasculature and myocardium, α-SMA–expressing myofibroblasts are thought of as the main source of collagens in the regulation of perivascular and interstitial fibrosis [9,14,15].

In a non-inflammatory and non-pathological steady-state condition, CCR2^−^ macrophages are located in the atrioventricular node and interact with neighboring cardiomyocytes, typically exhibiting anti-inflammatory M2 activity. These cells are derived from the embryonic yolk sac and fetal liver and have shown the potential to reduce the pro-inflammatory mediators release from cardiomyocytes and to inhibit the recruitment of monocyte-derived CCR2^+^ macrophages [88]. Furthermore, CCR2^−^ macrophages have also been shown to facilitate cardiac regeneration by increasing the proliferation and efferocytosis of apoptotic cells [89,92]. In contrast, an augmented influx of bone marrow-derived CCR2^+^ macrophages can replace part of the predominantly embryonic-derived CCR2^−^ macrophages, release a large number of cytokines, promote fibroblast proliferation, and cause cardiac interstitial fibrosis (Figure 4). We and others have previously reported that the expression of MCP-1 and the number of CCR2^+^ are increased in the arterial walls and myocardium in hypertensive animals. In the animal model of Ang II infusion or gene knockout, the pharmacologic blockade of the AT1R, specific ablation of macrophages, or abrogation of MCP-1 or CCR2^+^ markedly prevents the release of vascular inflammatory cytokines, attenuates the early development of myocardial fibrosis, and suppresses the progression of concentric to eccentric cardiac hypertrophy by reducing the numbers of non-resident CCR2^+^ macrophages [15,93,94,95].

## 6. The Roles of CCR2^−^ Resident Macrophages and CCR2^+^ Monocyte-Derived Macrophages in Ang II and Pressure Overload-Induced Heart Failure

Heart failure (HF), is not a disease but a multifaceted clinical syndrome, caused by a structural or functional impairment in the heart to fill with and pump blood, leading to a decline in stroke volume and cardiac output. HF affects approximately 6.7 million individuals in the United States with a high 5-year mortality rate after diagnosis [96]. Clinically, HF is categorized into two broad subtypes: left-sided HF and right-sided HF. In biventricular HF, also known as congestive HF, both sides of the heart are affected. Left-sided HF may be present with a preserved ejection fraction (HFpEF), defined by an EF of greater than 50%, and characterized by cardiac fibrosis that impairs cardiac diastolic relaxation and function. HFpEF currently accounts for approximately 50% of the HF population. HF in other patients is defined by a reduced ejection fraction (HFrEF) with an EF of less than 40% and characterized by cardiomyocyte cell death and chamber dilatation with reduced systolic function (Figure 5). A relatively new classification with an EF ranging from 41–49% is referred as a mid-range HF (HFmrEF). Clinical characteristics and outcomes often overlap in HFpEF and HFrEF [97,98].

### 6.1. The Role of CCR2^−^/CCR2^+^ Macrophages in HFpEF

In HFpEF, patients may experience a heterogeneous condition with many risk factors, such as hypertension, metabolic syndrome, valvular heart disease, arrhythmia, infection, diabetes, or age; however, approximately 60–80% of the patients with HFpEF are presented with hypertension. Chronic pressure overload induced by hypertension is considered as one of the most significant causes in the development of HFpEF [98,99]. Hypertension increases the workload on the heart, and induces structural and functional changes in the myocardium, including systemic inflammation, coronary endothelial dysfunction, increased arterial stiffness, tissue hypertrophy, perivascular and interstitial fibrosis, and cardiomyocyte impairment. All these changes lead to an increase in myocardial stiffness, a reduction in left ventricular (LV) diastolic compliance, and impaired cardiomyocyte relaxation. The pressure-volume relationship indicates an elevation in end-diastolic pressure along with a reduction LV end-diastolic volume and stroke volume. Therefore, hypertension and increased afterload are important drivers of cardiac remodeling in HFpEF [99,100].

The heart is composed of cardiomyocytes and non-myocytic cell types (fibroblasts, endothelial cells, and immune cells), which maintain physiological cardiac function via cellular crosstalk [100]. As previously discussed, the phenotype of CCR2^+^ macrophages is altered in abundance in response to pressure overload and play a critical role in the management and progression of pathological conditions, which can result in maladaptive cardiac remodeling and heart failure. Inflammation and reactive interstitial myocardial fibrosis are particularly important in intriguing the process of hypertensive cardiac remodeling in HFpEF. CCR2^+^ macrophages, once activated, release a variety of cytokines such as TGF-β, MCP-1, TNF-α, and IL-10, and cause excessive interstitial deposition of collagen proteins, leading to severe tissue stiffness, hypertrophy, and impaired cardiac diastolic function. In the myocardial biopsies obtained from patients with HFpEF and the mice model of aldosterone infusion, a high density of myocardial macrophages, due to the increased recruitment of monocytes from the bone marrow and the spleen, can be detected in parallel with a constellation of hematopoietic activation and diastolic dysfunction. Depletion of macrophages with antibody-based clearance [101], clodronate liposome [102], or depletion of IL-10 [103] has shown to inhibit inflammation and myocardial fibrosis and to improve echocardiographic indices of diastolic function, suggesting macrophage participation in adverse tissue remodeling and repression of cardiac function in HFpEF [104].

Cardiac macrophages can be differentiated according to their expression of CCR2 into resident CCR2^−^ and recruited CCR2^+^ subsets. CCR2^−^ macrophages are tissue-resident cells, and primarily located in the viable ventricular wall, coronary artery, and atrioventricular node to perform physiological functions such as endocytosis, angiogenesis, proliferation, regeneration, and cardiac electrical conduction (Figure 5) [12,105]. In murine cardiac tissue, CCR2^−^ macrophages constitute up to 5–10% of the non-myocyte population and play a major role in initiating the cardiac repair process in response to non-ischemia insults [12,106]. Previous studies involving mice models of cardiac pressure overload (e.g., transverse aortic constriction, TAC) and neonatal injury have shown that the blockade of CCR2^−^ macrophages using an antibody against macrophage colony-stimulating factor 1 receptor enhances cardiac fibrosis and blunts angiogenesis. Furthermore, knockout of CCR2^−^ macrophages shows enhanced recruitment of CCR2^+^ macrophages/neutrophils and aggravated fibrosis, suggesting a role of CCR2^−^ in the homeostasis and pathophysiology of cardiac remodeling in response to pressure overload [107,108,109].

CCR2^+^ macrophages are preferentially distributed in the trabecular projections of the endocardium and are embedded in collagen-rich scar tissue to promote inflammation, hypertrophy, and fibrosis. In the mouse model of pressure overload-induced heart failure with HFpEF, the CCR2^+^ macrophage population was increased via enhanced proliferation of bone marrow hematopoietic stem cells to replace the CCR2^−^ macrophages and participate in the pathogenesis of interstitial fibrosis, hypertrophy, and cardiac dysfunction. Therefore, the activation of CCR2^−^ macrophages early in the remodeling process may prevent the deterioration of cardiac function [110,111]. Increased production of MCP-1 and expression of intercellular adhesion molecules facilitates the migration of CCR2^+^ monocytes, activates fibroblast proliferation, promotes perivascular fibrosis, and increases cardiomyocyte stiffness and adverse remodeling in HFpEF (Figure 5) [91]. Accordingly, selective inhibition of CCR2^+^ macrophages was shown to inhibit left ventricular hypertrophy, ameliorate diastolic function, and reduce cardiac interstitial fibrosis and inflammation in HFpEF [112].

### 6.2. The Role of CCR2^−^/CCR2^+^ Macrophages in HFrEF

Common causes of HFrEF include myocardial infarction (MI), dilated cardiomyopathy, valvular heart disease, and cardiotoxicity. Clinically, HFrEF is predominantly associated with myocardial infarction (MI) [113]. The pathogenesis of post-MI in HFrEF is associated with multifactorial processes including oxidative stress, calcium dysregulation, and mitochondrial damage, all of which ultimately result in endothelial dysfunction, replacement fibrosis, cardiomyocyte cell death, and systolic dysfunction. Along with the infiltration of neutrophils and monocytes into the injured myocardium after MI, a significant release in pro-inflammatory chemokines and cytokines from these inflammatory cells is identified [114]. It has been previously demonstrated that cardiac CCR2^−^ macrophages located in the non-infarcted myocardium (remote zone) play a protective role following coronary artery occlusion (Figure 5). Inducible ablation of these macrophages using the Cx3cr1creERT2 system prior to ischemia limits cardiac adaptive response and aggravates cardiac dysfunction [115]. Furthermore, CCR2^−^ macrophages can regain their cardioprotective function and begin to remove necrotic tissue via phagocytosis and secretion of proteolytic enzyme after MI [116]. Studies have shown that the depletion of CCR2^−^ macrophages before MI results in significant changes in the myocardium, expressed as augmented CCR2^+^ macrophage proliferation, increased infarct area, exaggerated remodeling process, and reduced systolic function. Therefore, CCR2^−^ macrophages facilitate wound healing and cardiac regeneration by promoting myofibroblast proliferation, collagen deposition, and angiogenesis during the post-MI recovery period [31,117].

In the mouse model of dilated cardiomyopathy, the composition and dynamics of CCR2^−^ macrophages residing within the chronically failing heart have been identified. CCR2^−^ macrophages maintain adequate cardiac output by promoting compensative ventricular enlargement and coronary angiogenesis via a transient receptor potential vanilloid 4-dependent pathway and an upregulation of growth factor expression in CCR2^−^ macrophages [107]. These data suggest that resident CCR2^−^ macrophages represent a protective population, which mediates the adaptive remodeling and survival of the myocardium in the chronically failing heart. In contrast, a sequential recruitment of neutrophils and CCR2^+^ macrophages after MI, which is triggered by myelopoiesis in the bone marrow and spleen, is associated with the inflammatory sequelae, necrotic debris clearance, fibrotic tissue formation, and tissue injury exacerbation. The accumulation of CCR2^−^ and CCR2^+^ macrophages in the infarcted myocardium after MI depends upon the ischemic time. Within the first 24 h, CCR2^−^ macrophages are replaced by CCR2^+^ macrophages. The recruitment of CCR2^+^ macrophages to the acutely injured heart after MI induces cytokine and chemokine synthesis, promotes pro-inflammatory leukocyte recruitment, increases ventricular wall tension, and augments adverse remodeling. Accordingly, the depletion of CCR2^+^ macrophages post-MI prevents the recruitment of monocyte-derived macrophages circulating in blood and decreases the accumulation of type I IFN-biased macrophages that are implicated in adverse LV remodeling [118]. These results revealed the different roles of CCR2^−^ macrophages and CCR2^+^ macrophages in cardiac tissue repair and injury during maladaptive remodeling in HFrEF.

### 6.3. CCR2^+^ Macrophages and Ang II in the Transition of HFpEF to HFrEF

Abundant experimental evidence has highlighted the involvement of CCR2^+^ macrophages in Ang II-induced adverse remodeling and cardiac dysfunction, which potentially explain the mechanisms of action underlying the development of HFrEF. To demonstrate Ang II-induced cardiac remodeling through an interaction between inflammatory chemokine (CXCL1) and its receptor (CXCR2) on macrophage activation, a previous study in mice reported that Ang II infusion-induced cardiac hypertrophy, fibrosis, and inflammation are significantly attenuated when treated with CXCL1 neutralizing antibody, CXCR2 inhibitor SB265610, or CXCR2 knockout, suggesting a causative role of CXCL1-CXCR2 axis in the pathogenesis of Ang II-induced cardiac remodeling [119]. These data were consistent with the results showing a reduction in bone marrow-derived CXCR2^+^ monocyte infiltration in the myocardium and an inhibition of cardiomyocyte hypertrophy and myofibroblast differentiation in cultured cardiomyocytes [120]. Clinically, the increased abundance of cardiac CCR2^+^ macrophages is correlated with worsened cardiac systolic function and chamber dilatation in HF patients that undergo LV assist device implantation [120].

It has been previously reported that the progression of HFpEF to HFrEF is associated with alterations in the ECM [120]. The increased TGF-β and collagen contents released from CCR2^+^ macrophages and myofibroblasts suggest that there is an increased deposition of ECM within the hypertrophied heart. Since there are minimal changes in Ca^2+^ homeostasis and myofibril, the elevation of afterload during HFpEF results not only in cardiomyocyte hypertrophy, but also in changes in the ECM, suggesting that the development of HFpEF depends more on ECM turnover [121]. The most prominent hemodynamic findings in patients with HFpEF are related to elevated LV pressure secondary to incomplete myocardial relaxation and diastolic dysfunction, which can lead to an increase in left atrial peak pressure, stiffness, wall stress, and atrial fibrillation. Dysfunction of the left atrium exacerbated by this remodeling process eventually causes long-standing pulmonary hypertension and increases the risk of death in patients with HFpEF. Although patients with HFpEF have the same comorbidities as those with HFrEF, preclinical and clinical evidence suggests that HFpEF and HFrEF emerge from distinct pathophysiological processes. Metabolic and inflammatory alterations act in concert in cardiomyocytes and other cell types to provoke HFpEF, whereas HFrEF is characterized by perturbations of intracellular energy homeostasis, calcium handling, and cardiomyocyte loss [121,122].

### 6.4. The Role of CCR2^+^ Macrophages in Pressure Overload-Induced Cardiac Remodeling

Application of TAC (e.g., resembling cardiac stress under pressure overload) in animals has been widely used to identify the mechanisms of action underlying Ang II-induced adverse remodeling and cardiac injury, and to evaluate the pharmacological effects of drugs on cardioprotection. We and others have shown that TAC-induced pressure overload increases myocyte hypertrophy and fibrosis, and attenuates the cardiac function, which are all blocked by the AT1R antagonist, independent of the blood pressure-lowering effects [14,123]. These animal studies have provided abundant experimental evidence showing that Ang II is the most important trigger in the development of both adaptive and adverse remodeling in the early and late stages of pressure overload. The elevated levels of Ang II during the early stages are thought to induce adaptive effects such as cardiac concentric hypertrophy and to maintain cardiovascular homeostasis; however, circulation of Ang II in the bloodstream for a prolonged period of time increases the rate of monocyte seeding in the heart via mobilizing monocytes from the bone marrow and the spleen. An enhanced influx of monocytes into the heart can induce change in the composition of resident macrophages from predominantly embryonic-derived CCR2^−^ macrophages to monocyte-derived CCR2^+^ macrophages, leading to cardiac diastolic and systolic dysfunction and the progression of adverse concentric to eccentric hypertrophy and HF [124].

We have demonstrated that 4 weeks of Ang II infusion in rats induce hypertension and cardiac hypertrophy, which are associated with massive macrophage infiltration and perivascular and interstitial fibrosis as seen in pressure overload [9,14,15]. Clinically, ACEIs or ARBs have now become the preferred therapeutic drugs to treat patients with HFrEF. Experimental studies have already detailed the mechanisms of action underlying the protective effects of ACEIs and ARBs, including the inhibition of ROS, the reduction of intracellular Ca^2+^ overload, the downregulation of protein kinase C/MAPKs, and many other various signal-transduction pathways in HF [125,126]. Although ACEIs and ARBs have not shown a significant reduction in mortality and morbidity in patients with HFpEF [127,128], we have previously reported that macrophage accumulation, cardiac hypertrophy, and collagen deposition are significantly reduced with the use of AT1R antagonist, GLP-1 agonist, and curcumin (an anti-oxidant substance), which is consistent with the improved cardiac function in the rat models of Ang II infusion or TAC, suggesting that an interruption of Ang II/AT1 axis may delay the development of HFrEF [9,14,15].

### 6.5. Pressure Overload Enhanced Chemokine Production and CCR2^+^ Macrophage Activation

In the mouse model of TAC, the pressure overload results in the early upregulation of CCL2, CCL7, CCL12 chemokines, and CCR2^+^ infiltrating macrophages in the heart, along with an increase in the concentration of Ly6C^high^CCR2^+^ monocytes in the blood at 1 week post-TAC. This was accompanied by upregulation of the pro-inflammatory cytokine TNF-a and increased expression of TGF-β. Treatment with the selective spiropiperidine small molecule RS-504393, CCR2 receptor antagonist during TAC was found to inhibit CCR2^+^ macrophages and reduce pathological hypertrophy, fibrosis, and systolic dysfunction during the late phase of pressure overload [111].

CCR2^+^ macrophages are responsible for the recruitment of fibroblast precursors into the heart in pressure overload-induced cardiac fibrosis and dysfunction. Accordingly, CCR2^+^ knockout significantly reduced the accumulation of bone marrow-derived fibroblast precursors, as evidenced by reduced hematopoietic markers CD34 and CD45, and the levels of inflammatory mediators such as fibronectin, α-smooth muscle actin, and the mesenchymal marker (collagen I), in conjunction with a reduction in left ventricular dilatation and an improvement in cardiac systolic function [129]. In addition, genetic deletion of MCP-1 has been shown to reduce the induction of types I and III collagen, the level of TGF-β1, and the expression TNF mRNA, in consistent with a reduction in the CD34+/CD45+ fibroblast populations and an attenuation in hypertrophy and hypertension [130,131]. These results suggest that the tissue-remodeling responses engendered by macrophages in the pressure-overloaded heart are primarily linked to monocyte-derived Ly6C^high^CCR2^+^ macrophages, and that blockade of CCR2^+^ macrophage infiltration during the compensated phase may delay the transition to cardiac systolic dysfunction and failure as seen in HFrEF [130].

## 7. Future Perspectives

Pharmacological inhibition of Ang II-induced cardiac maladaptive remodeling has been conventionally used in the treatment of patients with HF. Recently, developing advanced methods for cardiac repair through modulating macrophages has attracted more attention.

### 7.1. Pharmacologically Targeted HF Treatment

The thresholds of LVEF continue to represent the fundamental parameter for therapeutic management. ACEIs, ARBs, β-blockers, and mineralocorticoid receptor antagonists remain the first-line therapy in patients with HFrEF, but are overall not effective in patients with HFpEF. Based on the analysis of most clinical characteristics, all-cause mortality, prognosis, and treatment responses to the drugs, HFmrEF is overall more close to HFrEF than to HFpEF, and might therefore be interpreted as being a mild form of HFrEF [121,132]. Recently, sodium glucose cotransporter 2 (SGLT2) inhibitors have been selected to treat patients with HFpEF, but also with HFmrEF. SGLT2 inhibitors are a class of medications that inhibit sodium-glucose transport proteins in the nephron. Aside from adequate control of blood sugar and blood pressure, clinical data in patients with HFpEF have shown that the SGLT2 inhibitors reduce worsening heart failure, cardiovascular death, and transition from preserved to reduced EF as a result of an acute ischemic event [121,133,134]. From animal studies, SGLT2 inhibitors have shown the beneficial effects on attenuation of inflammation, improvement of endothelial function, and suppression of collagen synthesis and fibrosis [135,136]. In this regard, we have reported that GLP-1 agonist liraglutide, another glucose-lowering drug, is effective in the alleviation of hypertension, hypertrophy, and cardiac fibrosis irrespective of diabetes status [137]. The level of Ang II is increased in heart failure patients depending upon the time course in pathogenesis. Therefore, targeting the components of Ang II may produce significant benefits when this mechanism turns maladaptive in HFmrEF and HFpEF. Although there is a substantial gap in the treatment of patients with HFpEF using ACEIs and ARBs [138], studies from our laboratory and others have shown that Ang II-induced cardiac hypertrophy and fibrosis are significantly inhibited by AT1R antagonists. Thus, more basic, translational, and clinical research are needed to delineate whether the dual antagonism of multiple pathological fibrotic pathways with ACEIs or ARBs plus SGLT2 inhibitor or GLP-1 receptor agonist might result in a synergistic effect and confer greater benefits in patients with HFpEF and HFrEF.

Ang II promotes cardiac dysfunction and failure by differentiating monocyte-derived CCR2^+^ macrophages [129,131], which can be blocked by pharmacological targeting of the CCL2/CCR2 axis [139,140]. Blockade of upstream signaling using ACEIs or ARBs exhibits cardioprotection at a compatible level as those in anti-macrophage therapy [141]. Therefore, further studies are warranted that simultaneous application of ACEI or ARB combined with other existing anti-macrophage agents might produce additive effects against Ang II-induced maladaptive cardiac remodeling and heart failure. Collectively, it is important to investigate the time course when and how macrophages are involved in the progression of HFpEF to HFrEF in animal studies. Taken together, these potential experimental investigations may open another therapeutic window to explore whether combined medication could enhance efficacy in the treatment of patients with HFpEF, HFmrEF, or HFrEF [142,143].

### 7.2. Polarization of Resident Macrophages and CCR2^+^ Macrophage-Targeted HF Treatment

As discussed above, the phenotypic switch of resident macrophages from M2- to M1-like and the recruitment of monocyte-derived macrophages play a key role in maladaptive cardiac repair and HF. Both types of macrophages interact with other non-cardiac cells such as fibroblasts to control the progression of hypertension, hypertrophy, and fibrosis. Different preclinical strategies targeting the improvement of cardiac repair by attenuating phenotypic switch and recruitment of CCR2^+^ macrophages have been well reviewed recently [144], including the depletion of CCR2^+^ macrophages using clodronate liposomes, the disruption of chemokine CCL5 and CCL2 using receptor antagonists or inhibitors, the inhibition of macrophage migratory inhibitory factor, and the silencing of CCR2 mRNA. Transplantation of M2 macrophages after cardiac ischemia and reperfusion significantly reduced tissue fibrosis and prevented adverse remodeling. Furthermore, injection of autologous M2 macrophages in patients has been shown to improve neurological recovery without serious adverse events [144]. However, it has also been reported that administration of autologous bone marrow macrophages via transendocardial injection shows no improvement in cardiac function. It remains challenging by targeting the use of anti-macrophages in the treatment of various cardiovascular diseases [145,146]. The questions that still need to be addressed are: (1) when macrophage-targeted drugs should be delivered because the development of cardiovascular diseases is a complex chronic process, and the time course of the treatment in acute vs. chronic cardiovascular events should be illustrated; (2) the dose and route of delivering anti-macrophage drugs should be determined; (3) to make the use of therapeutic macrophages for treating cardiovascular diseases via nanoparticles or cell transplantation, the efficacy and safety of such a treatment needs to be comprehensively tested in patients. Currently, although it is still questionable in the translation of macrophage-targeted therapies from basic findings to clinical treatments, the evidence obtained from animal studies in the investigation of suppressing inflammation and macrophages with the therapeutic approach may support the opening of a new era in the improvement of cardiac maladaptive remodeling and failure in patients.

## 8. Conclusions

In this review, we summarize the latest reports in the understanding of Ang II/AT1R-induced cardiac microvascular endothelial dysfunction, hypertrophy, and heart failure, primarily aiming at macrophages-mediated signaling pathways. Macrophages based on their origin, phenotypes, and localization in cardiac tissue can play a beneficial and/or detrimental role in healthy hearts and in cardiovascular diseases. Study approaches using specific macrophage gene knockout or macrophage ablation have provided recent experimental evidence showing that macrophages act as a central contributor in hypertension, cardiac hypertrophy, and failure. Inhibition of macrophage-mediated inflammation, microvascular dysfunction, and heart failure using ACEIs, ARBs, or other anti-macrophage interventions offer an immense opportunity for therapeutic development. Although abundant progress in animal studies has been made to illustrate the pivotal role of macrophages in Ang II-mediated cardiovascular dysfunction, translating these findings from the experimental findings into clinical practice needs additional investigation, particularly focusing on mechanisms of protective action underlying the inhibition of Ang II/AT1R/macrophage axis-induced cardiac pathologies, such as hypertension, maladaptive cardiac repair, fibrosis, and heart failure progression.

## 9. Limitations

This review primarily discusses the various roles of macrophages in Ang II-induced hypertension, cardiac vascular dysfunction, hypertrophy, fibrosis, and heart failure, as understood in recent years. Although we have extensively read the literatures, some of the latest experimental outcome-based studies may not be fully discussed in this review due to the rapid progress of molecular/cellular research and the emerging technologic approaches in this field.

## Figures and Tables

**Figure 1 cells-13-02001-f001:**
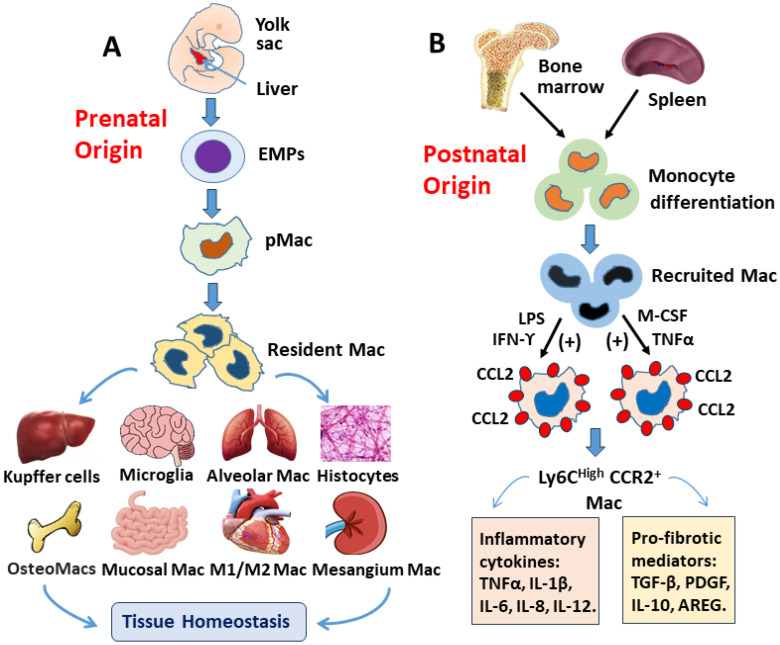
Origin and development of tissue-resident and monocyte-derived macrophages. During the prenatal period (**A**), the tissue-resident macrophages (Mac) are generated during embryonic development through self-renewal proliferation in the yolk sac and fetal liver from erythro-myeloid progenitors (EMPs) and pre-macrophages (pMac), which can differentiate into long-lived tissue-resident macrophages. Distinct macrophage subpopulations crosstalk with specialized tissue cells, support tissue function, and maintain homeostasis during steady-state adulthood. During the early postnatal stage (**B**), bone marrow-derived hematopoietic stem cells (HSCs) give rise to short-lived and long-lived macrophages through a series of sequential differentiation. Conventionally, native macrophages, stimulated by cytokine-like lipopolysaccharides (LPS), interferon (IFN), colony-stimulating factor (CSF), and tumor necrosis factor (TNF), are polarized into Ly6c^high^CCR2^+^ macrophages and release pro-inflammatory cytokines such as TNF-α, IL-1β, IL-6, IL-8, IL-12, and pro-fibrotic mediators TGF-β, IL-10, platelet derived growth factor (PDGF), and amphiregulin (AREG), which drive cell infiltration and tissue regeneration/remodeling.

**Figure 2 cells-13-02001-f002:**
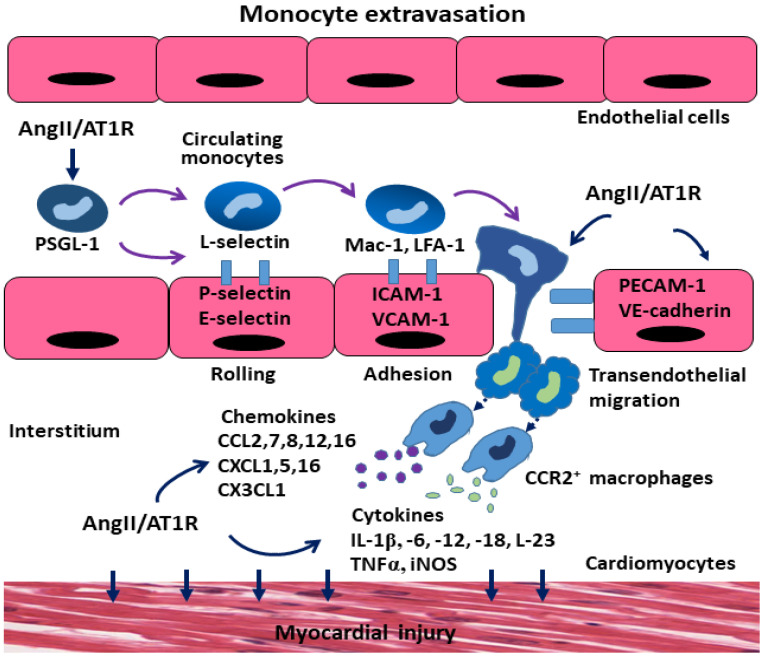
The pathological mechanisms underlying monocyte extravasation with angiotensin II (Ang II) stimulation. Transendothelial migration of monocytes into the interstitium involves the sequential interaction of distinct receptors on the surface of monocytes and endothelial cells. Ang II via the AT1 receptor upregulates the level of P-selectin glycoprotein ligand-1 (PSGL-1) on monocytes to mediate initial contact (rolling) between circulating monocytes and the vascular endothelium via upregulating lectin-like molecules (L-selectin) on monocytes and (P-selectin, E-selectin) on the endothelium. Subsequent monocyte firm adhesion to the endothelium relies on binding of the intercellular adhesion molecule-1 (ICAM-1) and the vascular cell adhesion molecule-1 (VCAM-1) to their ligands, including macrophage antigen 1 (Mac-1; CD11b/CD18) and lymphocyte function-associated antigen 1 (LFA-1; CD11a/CD18). This is followed by transendothelial migration, which is facilitated by additional Ig-supergene family member expression, including endothelial platelet-endothelial cell adhesion molecule (PECAM-1) and vascular endothelial (VE)-cadherin. Monocyte-derived C-C chemokine receptor 2 (CCR2^+^) macrophages in the interstitium induce myocardial injury by releasing various chemokines and cytokines.

**Figure 3 cells-13-02001-f003:**
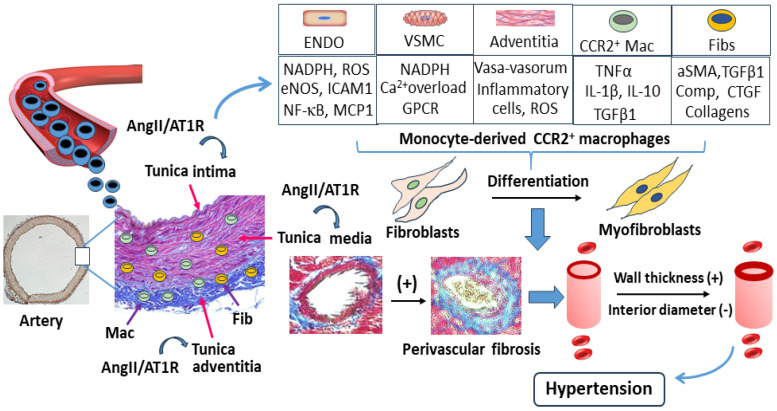
Schematic overview of angiotensin II (Ang II)-induced perivascular fibrosis and hypertension. Moving from left to right, depicting that the infiltration of monocyte-derived CCR2^+^ macrophages in the aortic wall is a hallmark of Ang II-mediated vascular injury. On endothelium (Endo), Ang II facilitates transmigration of CCL2-expressed CCR2^+^ macrophages by stimulating inflammatory responses, adhesion molecules (ICAMs), nuclear factor kappa light-chain-enhancer of activated B cells (NF-κB), and monocyte chemoattractant protein-1 (MCP-1). In vascular smooth muscle cells (VSMC), the vessel wall degradation and remodeling by Ang II are largely mediated through the production of NADPH oxidase, Ca^2+^ overload, and the activation of G-protein couple receptor (GPCR) signaling. In adventitia, Ang II facilitates the infiltration of CCR2^+^ macrophages by vasa vasorum neovascularization and stimulates the interaction between adventitial and perivascular tissues. Macrophages largely participate in vessel wall degradation and remodeling through the production of pro-inflammatory cytokines such as TNF-α, IL-1β, IL-10, and TGF-β1. Proliferation of fibroblasts (Fibs) to alpha smooth muscle actin (αSMA)-expressing myofibroblasts through the secretion of TGF-β from CCR2^+^ macrophages cause the release of collagen assembly-related extracellular matrix proteins including cartilage oligomeric matrix protein (Comp), connective tissue growth factor (CTGF), and collagens, leading to perivascular fibrosis and hypertension through this maladaptive vascular remodeling process.

**Figure 4 cells-13-02001-f004:**
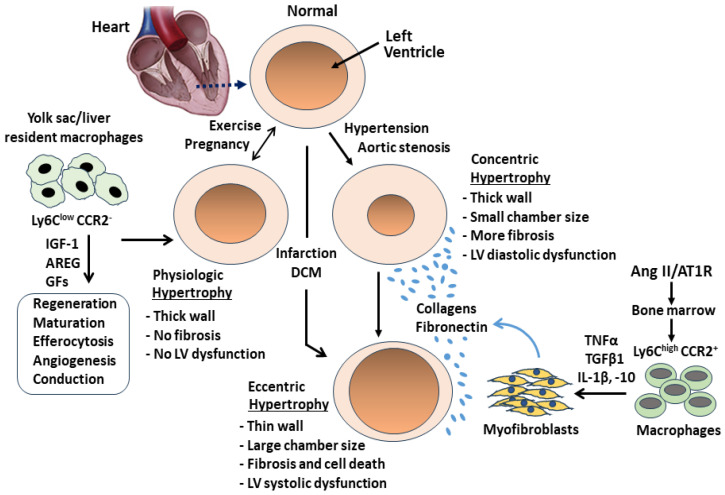
Diagrammatic representation of physiological and pathological hypertrophic responses in the heart. In general, hypertrophy results in an increase in cardiomyocyte size and tissue mass depending upon physiological and pathophysiological stimuli. In exercise- and pregnancy-induced reversible physiological hypertrophy, the heart undergoes an individual cardiomyocyte increase in width with normal left ventricle (LV) cardiac function. In the physiological condition, cardiac resident Ly6C^low^CCR2^−^ macrophages from the embryo act as cellular chaperones for tissue homeostasis via insulin-like growth factor-1 (IGF-1), amphiregulin (AREG), and growth factors (GFs). Pathological hypertrophy can be classified into pressure or volume overload (e.g., concentric vs. eccentric hypertrophy). Development of concentric hypertrophy is associated with hypertension and aortic stenosis, characterized as wall and septal thickening, a loss of chamber area, tissue fibrosis, and LV diastolic dysfunction. Over time, this state can deteriorate into dilated and eccentric hypertrophy, expressed as wall thinning, chamber dilatation, and an increase in wall tension along with LV systolic dysfunction. Some other disease states, such as myocardial infarction and dilated cardiomyopathy (DCM), can lead directly to eccentric hypertrophy without a prior concentric remodeling phase. In Ang II/AT1R-mediated concentric and eccentric hypertrophy, monocyte-derived Ly6C^high^ CCR2^+^ macrophages from the bone marrow promote the proliferation of myofibroblasts and lead to cardiac tissue fibrosis by releasing cytokines and depositing collagens and fibronectins.

**Figure 5 cells-13-02001-f005:**
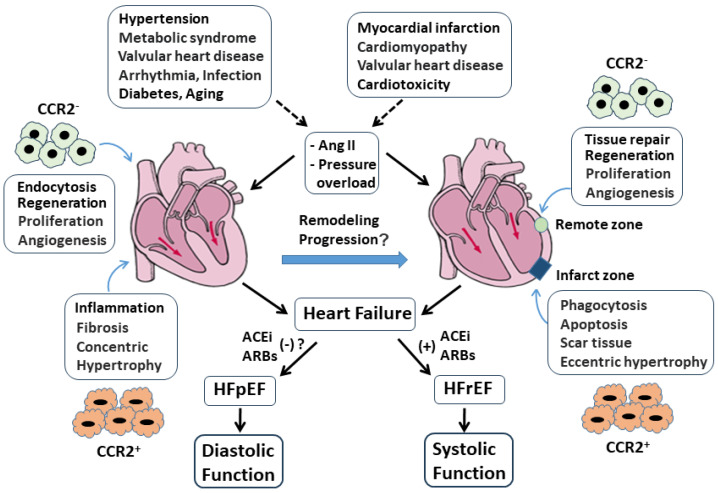
Role of macrophages in Ang II and pressure overload induced adverse cardiac remodeling and failure. A multifaceted pathological stimulation increases Ang II level and pressure overload, results in maladaptive remodeling, and leads to heart failure with a preserved ejection fraction (HFpEF) and diastolic dysfunction or a reduced ejection fraction (HFrEF) and systolic dysfunction. In the adaptive stage or the remote zone after myocardial infarction (MI), the tissue C-C chemokine receptor (CCR2^−^) resident macrophages maintain cardiac homeostasis, proliferation, and angiogenesis, promote tissue repair and regeneration, and preserve cardiac function, whereas in the conditions of concentric hypertrophy mediated by Ang II stimulation or pressure overload, as well as eccentric hypertrophy after MI, the replacement of CCR2^−^ resident macrophages by recruited CCR2^+^ macrophages in the myocardium or infarct zone evoke inflammation, induce fibrosis and scar formation, and deteriorate cardiac function. Depending upon the alterations in the degree of cardiac remodeling, e.g., extracellular matrix deposition, heart failure with HFpEF may develop to the stage of HFrEF. Although there is a controversial issue regarding the treatment of patients with HFpEF using angiotensin-converting enzyme inhibitors (ACEIs) or angiotensin receptor blockers (ARBs), both drugs significantly inhibit tissue fibrosis, preserve cardiac function, and reduce mortality in patients with HFrEF.

**Table 1 cells-13-02001-t001:** Chemokines, cytokines, adhesion molecules and cell types involved in the extravasation of monocyte-derived macrophages.

Ligands	Receptors	Cell Type Involved	Function
**Chemokines** CCL2 (MCP-1), CCL8 (MCP2), CCL7 (MCP-3), CCL13 (MCP-4), CCL16 (HCC4) CCL-3 (MIP-1α), CCL4 (MIP-1β), CCL11 (eotaxin), CCL14 (HCC1), CCL16 (HCC4) CXCL6 (GCP2), CXCL8 (IL-8) CXCL1 (GROα), CXCL2 (GROβ), CXCL3 (GROϒ), CXCL5 (ENA-78), CXCL6 (GcP2) CX3CL1 (fractalkine)	CCR2, CCR3, CCR5 CCR1, CCR5 CXCR1 CXCR2 CX3CR1	Monocytes, endothelial cells, T-cells, leukocytes Monocytes, endothelial cells, T-cells, leukocytes Monocytes, neutrophils Monocytes, endothelial cells neutrophils Monocytes, T-cells, endothelial cells	Inflammatory monocyte trafficking Macrophage and NK-cell migration Monocyte and NK-cell migration T-cell-DC interaction Neutrophil trafficking Neutrophil trafficking Neutrophil trafficking Monocyte, NK, and T-cell migration
**Cytokines** IL-1β IL-6 IL-12, IL-23 TNF-α, iNOS	CD121a, CDw121b CD126, 130 CD212 CD120a, b	Monocytes/macrophages B and T cells, monocytes, endothelial cells, fibroblasts Phagocytic cells, microglial and dendritic cells Macrophages, mast cells, NK cells, T and B cells	Mediator of systemic effects of IL-6 Increase recruitment of monocytes to the inflammation site Pro-inflammatory neutrophil TNF-α, IFN-ϒ synthesis Increase in permeability, Stimulation of adhesion molecules
**Ligands/adhesion molecules** PSGL-1 PSGL-1 MCP-1 ICAM1/CD54 ICAM1/CD54 VCAM-1/CD106 PECAM1 (CD31)	**Integrins/receptors** P-selectin (CD62P), E-selectiin (CD62E) L-selectin CCR2 Mac-1 (αMβ2-integrin or CD11b-CD18) LFA-1 (αLβ2-integrin or CD11a-CD18) VLA-4 (α4β1-integrin or CD49d-CD29b) PECAM1	Endothelial cells Monocytes Monocytes, endothelial cells Endothelial cells Endothelial cells T cells and monocytes Endothelial cells Endothelial cells Monocytes, endothelial cells	Monocyte rolling Monocyte rolling Monocyte chemotaxis Cell adhesion/transmigration T cell activation, cell transmigration Cell adhesion/transmigration Cell adhesion/transmigration Cell adhesion/transmigration

Abbreviations: CCL, C-C motif ligand; CXCL, CXC chemokine ligands; CX3CL1, chemokine (C-X3-C motif) ligand; TNF-α, tumor necrosis factor; iNOS, inducible nitric oxide synthase; PSGL-1, P-selectin glycoprotein ligand-1; L-selectin, lectin-like molecules; Mac-1, the macrophage-antigen-1; LFA-1, lymphocyte function-associated antigen 1; CCR, C-C chemokine receptor; MCP-1, monocyte chemoattractant protein-1; VLA-4, neutrophil β1-integrin very late antigen-4; ICAM1, intercellular adhesion molecule-1; VCAM-1, vascular cell adhesion molecule-1; NK, natural killer.

## Data Availability

No new data were provided in this review. Data sharing is not applicable to this article.

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
