# Peer review of "Pathophysiology of Angiotensin II-Mediated Hypertension, Cardiac Hypertrophy, and Failure: A Perspective from Macrophages"

_cells, 2024, doi:10.3390/cells13232001_

Round 1
Reviewer 1 Report
Comments and Suggestions for Authors
The review explores the role of macrophages in Angiotensin II (Ang II)-mediated cardiac hypertrophy, fibrosis, and heart failure. Ang II activates macrophages via the AT1 receptor, triggering inflammation and adverse cardiac remodeling. It highlights the balance between M1 (pro-inflammatory) and M2 (anti-inflammatory) macrophages in cardiac injury and homeostasis. Ang II also promotes the recruitment of monocyte-derived macrophages, worsening fibrosis and dysfunction. Additionally, the review discusses how Ang II enhances adhesion molecule expression, facilitating monocyte infiltration into the heart. The findings emphasize potential therapeutic strategies to reduce macrophage-driven inflammation and tissue remodeling in cardiovascular diseases. The manuscript was well-written, however, there are many areas to be improved, for examples,
1. Consider using clearer subheadings to delineate sections on mechanisms, experimental evidence, and therapeutic implications. This would enhance readability and allow for easier navigation through the complex information presented.
2. While the review covers significant findings related to Ang II and macrophages, it would benefit from a more extensive discussion of recent studies, particularly those published after the initial literature review. This could include emerging therapies targeting macrophage activity and their potential in clinical settings, which would provide a more rounded perspective on current research trends and gaps in knowledge.
3. The manuscript references various studies that support the role of macrophages in Ang II-induced cardiac remodeling. However, it would be beneficial to include more detailed descriptions of key experimental methodologies and results. For instance, discussing specific animal models used in studies could provide context for the findings and their applicability to human conditions.
4. The discussion on therapeutic strategies is somewhat limited. Expanding this section to include a broader range of potential interventions, such as the use of combination therapies (e.g., ACE inhibitors, ARBs, and anti-inflammatory agents), would be valuable. Additionally, discussing the challenges and limitations of translating these findings into clinical practice would provide a more balanced view of the current state of research.
5. The manuscript briefly mentions future perspectives but could elaborate on specific research questions that remain unanswered. For example, exploring the timing and mechanisms of macrophage involvement in the transition from heart failure with preserved ejection fraction (HFpEF) to heart failure with reduced ejection fraction (HFrEF) could be a critical area for future studies.
6. The manuscript mentions limitations in the current understanding of macrophage roles but could benefit from a more thorough discussion of the methodological challenges faced in this field of research. Addressing these challenges could provide insights into how future studies might overcome them.
7. The conclusion summarizes the findings well but could be strengthened by reiterating the clinical significance of the research. Emphasizing how these insights could lead to improved patient outcomes or novel therapeutic strategies would underscore the importance of the work presented.
Author Response
We very much appreciate the reviewer’s effort and time in reviewing this manuscript.
Response to Reviewer #1
The review explores the role of macrophages in Angiotensin II (Ang II)-mediated cardiac hypertrophy, fibrosis, and heart failure. Ang II activates macrophages via the AT1 receptor, triggering inflammation and adverse cardiac remodeling. It highlights the balance between M1 (pro-inflammatory) and M2 (anti-inflammatory) macrophages in cardiac injury and homeostasis. Ang II also promotes the recruitment of monocyte-derived macrophages, worsening fibrosis and dysfunction. Additionally, the review discusses how Ang II enhances adhesion molecule expression, facilitating monocyte infiltration into the heart. The findings emphasize potential therapeutic strategies to reduce macrophage-driven inflammation and tissue remodeling in cardiovascular diseases. The manuscript was well-written, however, there are many areas to be improved, for examples,
Comment 1: Consider using clearer subheadings to delineate sections on mechanisms, experimental evidence, and therapeutic implications. This would enhance readability and allow for easier navigation through the complex information presented.
Response 1: This is an excellent suggestion. We have added several subheadings highlighted as red in the text to enhance readability.
Comment 2: While the review covers significant findings related to Ang II and macrophages, it would benefit from a more extensive discussion of recent studies, particularly those published after the initial literature review. This could include emerging therapies targeting macrophage activity and their potential in clinical settings, which would provide a more rounded perspective on current research trends and gaps in knowledge.
Response 2: We modified a paragraph regarding the role of Ang II in the activation of macrophages and cardiac remodeling on page 21 and also a paragraph of macrophage-targeted therapy on page 21. Thank you!
Comment 3: The manuscript references various studies that support the role of macrophages in Ang II-induced cardiac remodeling. However, it would be beneficial to include more detailed descriptions of key experimental methodologies and results. For instance, discussing specific animal models used in studies could provide context for the findings and their applicability to human conditions.
Response 3: We have re-addressed the role of animal experimental models (i.e., pressure overload) in cardiac injury through Ang II-mediate pathways in paragraph 6.4 on page 19, 20. Thank you!
Comment 4: The discussion on therapeutic strategies is somewhat limited. Expanding this section to include a broader range of potential interventions, such as the use of combination therapies (e.g., ACE inhibitors, ARBs, and anti-inflammatory agents), would be valuable. Additionally, discussing the challenges and limitations of translating these findings into clinical practice would provide a more balanced view of the current state of research.
Response 4: The reviewer pointed out an important issue regarding how to translate the animal findings to clinical practice. This has been always concerned by the majority of investigators including our lab. For example, one important finding firstly reported from our lab (i.e., ischemic postconditioning, Zhao et al, Am J Physiol Heart Circ Physiol. 2003, 285:H579-88) has been widely cited and tested in patients. We agreed that it takes a long time to translate basic findings to clinical. We re-modified the potential use of combination therapies in maladaptive cardiac remodeling induced heart failure on page 21.
Comment 5: The manuscript briefly mentions future perspectives but could elaborate on specific research questions that remain unanswered. For example, exploring the timing and mechanisms of macrophage involvement in the transition from heart failure with preserved ejection fraction (HFpEF) to heart failure with reduced ejection fraction (HFrEF) could be a critical area for future studies.
Response 5: We agreed this comment, and put more emphasis on this discussion on page 22.
Comment 6: The manuscript mentions limitations in the current understanding of macrophage roles but could benefit from a more thorough discussion of the methodological challenges faced in this field of research. Addressing these challenges could provide insights into how future studies might overcome them.
Response 6: To target macrophage therapy in cardiovascular diseases, we have re-organized the discussion in the section of future perspectives based on newest references (i.e., Ref. 145-147), and pointed out the potential options of using drugs and cell transplantation on page 21-22.
Comment 7: The conclusion summarizes the findings well but could be strengthened by reiterating the clinical significance of the research. Emphasizing how these insights could lead to improved patient outcomes or novel therapeutic strategies would underscore the importance of the work presented.
Response 7: As suggested, we have reiterated the role of research in the improvement of patient’s treatment in a paragraph of conclusion on page 22.
Reviewer 2 Report
Comments and Suggestions for Authors
The review manuscript described Ang II-mediated cardiac hypertrophy, fibrosis and failure mainly concerning macrophages, which is very interesting and informative. I have two comments. Line 100, the authors mentioned that macrophages led to “aberrant repair”. In my opinion, macrophages have two roles, destructive and reparative. Therefore, macrophages may lead malfunction but also lead to be curative. Inflammation is necessary in the early stage of heart dysfunction, to clean the damaged cells majorly. Without this, the damage can be more aggravated. On the other hand, too exceeded inflammation may cause more destruction. I think it is better to describe the 2 sides of macrophage function, or explain why the authors described that macrophages can lead only “aberrant repair”. The other comment, Line 117, I do not think osteoblasts are generally recognized as the “macrophages” in the bone.
Author Response
We very much appreciate the reviewer’s effort and time in reviewing this manuscript.
Response to Reviewer #2
The review manuscript described Ang II-mediated cardiac hypertrophy, fibrosis and failure mainly concerning macrophages, which is very interesting and informative. I have two comments.
Comment 1: Line 100, the authors mentioned that macrophages led to “aberrant repair”. In my opinion, macrophages have two roles, destructive and reparative. Therefore, macrophages may lead malfunction but also lead to be curative. Inflammation is necessary in the early stage of heart dysfunction, to clean the damaged cells majorly. Without this, the damage can be more aggravated. On the other hand, too exceeded inflammation may cause more destruction. I think it is better to describe the 2 sides of macrophage function, or explain why the authors described that macrophages can lead only “aberrant repair”.
Response 1:
The reviewer is right. To make clear the role of macrophages in tissue remodeling, we deleted “aberrant” and changed it as “macrophages also lead to the maladaptive regeneration, repair and tissue fibrosis in line 101”. Furthermore, we also added CCR2+ cells from the bone marrow and spleen in the participation of inflammation, proliferation and maturation acting as M1-like resident macrophages during tissue remodeling in line 79.
.
Comment 2: Line 117, I do not think osteoblasts are generally recognized as the “macrophages” in the bone.
Response 2:
Thank you for your comment. It is true that osteoblasts are no longer recognized as the “macrophages” in the bone. We updated OsteoMacs in the bone (see in revised figure 1 and line 118) based on descriptions from other reviews. (Ref. Oscar Iglesias-Velazquez , et al. OsteoMac: A new player on the bone biology scene. Ann Anat. 2024, 254:152244. doi: 10.1016/j.aanat.2024.152244).

Round 2
Reviewer 2 Report
Comments and Suggestions for Authors
I have no further comments.